# Vulnerability and Agency in the Time of COVID-19: The Narratives of Child and Youth Care Workers in South Africa

**DOI:** 10.3390/ijerph20065010

**Published:** 2023-03-12

**Authors:** Andile Samkele Masuku, Reggiswindis Thobile Hlengwa, Lindelwa Vernon Mkhize, Maureen Nokuthula Sibiya

**Affiliations:** 1Student Governance and Development Unit, Student Services, Steve Biko Campus, Durban University of Technology, Durban 4000, South Africa; 2Department of Community Health Studies, Faculty of Health Sciences, Ritson Campus, Durban University of Technology, Durban 4000, South Africa; 3International Education and Partnerships, ML Sultan Campus, Durban University of Technology, Durban 4000, South Africa; 4Division of Research, Innovation and Engagement, Umlazi Campus, Mangosuthu University of Technology, Umlazi 4031, South Africa

**Keywords:** agency, COVID-19, child and youth care workers, mental health, vulnerability

## Abstract

In this paper, we use data generated through one-on-one interviews with 12 purposively sampled Child and Youth Care Workers to examine their narratives of work and life-related vulnerabilities and agency during the peak of the COVID-19 global pandemic in KwaZulu-Natal, South Africa. Our findings show that Child and Youth Care Workers were vulnerable to poor mental health. Indeed, working and socialising during the height of COVID-19 posed a mental toll on the Child and Youth Care Workers in this study, who experienced fear, uncertainty, anxiety and stress. Moreover, these workers faced challenges with working under the so-called new normal, which was instituted as part of a non-pharmaceutical response to slow and curb the spread of COVID-19. Finally, our findings show that Child and Youth Care Workers actively identified and applied specific emotionally-focused and physically-focused coping mechanisms to deal with the burden brought on by the pandemic. The study has implications for CYCWs working during crisis periods.

## 1. Introduction

Countries across the globe have witnessed and are still struggling with the battle of devastating health, economic and social crises caused by the novel coronavirus disease (COVID-19) [1]. COVID-19 is an infectious disease caused by severe acute respiratory syndrome coronavirus 2 (SARS-CoV-2) [2]. This rapidly spreading respiratory virus was first identified in the Chinese Province of Wuhan in November 2019 and quickly spread globally, eventually leading the World Health Organisation (WHO) to declare it a global health pandemic. The nature of its spread has posed significant challenges to public health and compromised the provision of social services across nations [3]. Further, this pandemic has revealed the vulnerability of our institutions and systems of governance regarding the economy, and health systems, including social spheres [4]. In March 2020, South Africa reported its first COVID-19 case, which led to nationwide terror and alarm [5]. In the aftermath of this first COVID-19 case, the country’s Department of Health (DoH) began to report an exponential increase in the number of identified positive cases, with significant household and community-level transmission. In response to the rapid increase in reported cases, the South African government, under the leadership of its president, Cyril Ramaphosa, declared a state of disaster under which a national lockdown was instituted [6]. The lockdown saw the suspension of all so-called non-essential activities, including a ban on local and international travel, the closure of educational institutions, and the introduction of curfews. However, essential services, such as the provision of healthcare and social services, were exempted from the ban.

COVID-19 has placed an increased demand on health and social care services with severe pressures faced by workers [7]. The pandemic has deeply changed the social and working environments in numerous ways [8]. COVID-19 has impacted our lives from financial to social interactions, family to businesses and physical to mental health, especially workers who have been at the frontline in fighting the pandemic [9]. The role played by social services in providing essential services to vulnerable populations during the height of the COVID-19 pandemic has been acknowledged [10]. Among these key actors are Child and Youth Care Workers (CYCW), who provide important services, such as responding to the needs of vulnerable children and youth, including those who are orphaned or considered at risk for social ills [11]. However, the literature remains mum about how CYCW negotiated their lives and worked during the height of the COVID-19 pandemic. In previous health-related pandemics, the work of CYCW was noted as positive and beneficial to their clientele. For example, at the height of the HIV and AIDS crisis that killed a number of parents and left behind a generation of orphaned children, CYCW responded by providing care and support to these vulnerable children [12]. Consequently, it was noted that the work provided by CYCW during the HIV and AIDS crisis became invaluable through, for example, nurturing agency among orphaned children [13]. Since COVID-19 is a newly discovered virus that has quickly reached pandemic levels [14], it becomes imperative to listen to the narratives of vulnerability and coping among frontline workers such as CYCW. “Vulnerability is a key concept to understanding risk exposure and outcomes associated with disasters and hazards” [15]. Whereas agency is the human capability to influence one’s functioning and the course of events through one’s actions [16].

Therefore, this paper examines narratives of vulnerability and agency among CYCW during the height of COVID-19 in South Africa.

## 2. Child and Youth Care Work in a Time of Crisis

COVID-19 has undeniably caused innumerable socioeconomic vulnerabilities throughout society [17]. For example, the pandemic has rendered children vulnerable to hunger, neglect and abuse, among other things, due to the instituted lockdowns that facilitated the closure of schools, further isolating and impacting children significantly [18]. COVID-19 has led numerous children, especially in poor and developing countries, to be exposed to the dangers of the street because of the deprivation of their right to education [19]. Furthermore, it has taken an immense toll on youth and families, especially those living in distressed households and communities that are characterised by poverty, limited resources, high unemployment rates, and violence [20]. COVID-19 has disrupted the delivery and use of routine services that are essential for sustaining livelihoods [21]. As this pandemic persists, it has been reported that several individuals are experiencing circumstances which lead to poor mental health performance, i.e., such as isolation, stress, anxiety, depression, anger, confusion and unemployment [22]. Hence, a call has been made for changes in public policies and health policies for the public health institutions to be prepared to deal with emergencies such as the one created by the pandemic [23]. Thus, a well-supported, appropriately equipped, well-informed, and protected social service workforce is important for curbing the negative effects resulting from the COVID-19 pandemic [24]. The work provided by CYCW is important in the context of health pandemics. These individuals’ work extends to spaces where young people live and learn to promote childhood development through daily planned activities [11]. The CYCW are responsible for nurturing effective relationships with children and youth as they interact with them every day [6]. Moreover, CYCW works with parents/caregivers and families to negotiate a viable social environment for children and youth that are regarded as vulnerable. Yet, as studies show, the COVID-19 pandemic has disrupted social life, including social spaces and people’s daily life routines [25]. The pandemic has also affected mental health by rendering individuals anxious and stressed [26]. A number of people across the world are suffering from the pandemic. Thus, a wave of mental health problems emerged [27]. This current health crisis points to a need for professionals who might help individuals to prepare for extreme and emotionally daunting conditions, especially during health-related pandemics [28].

It was argued that COVID-19 had impacted the way people live their lives, including how much time they spend together with others [29]. Indeed, there have been notable difficulties in maintaining and negotiating close relationships since COVID-19 disrupted such relationships by forcing people into social/physical distancing practices and isolation in cases where individuals test positive for the virus [30]. In such situations, CYCW become important gap fillers who continue to foster sustained relationships with young people and their families [31] in spite of the fact that COVID-19 has reconfigured how people engage [32].

## 3. Materials and Methods

### 3.1. Study Site and Context

This paper emerges from a larger qualitative study that examined narratives of vulnerability and agency among a select group of CYCW during the COVID-19 pandemic. The research sites were two residential Child and Youth Care Centres (CYCC) in the eThekwini Municipality, in the KwaZulu-Natal (KZN) province of South Africa. KZN remains the second hardest-hit province, after Gauteng, in terms of positive COVID-19 cases. In this province, as [33] revealed at the onset of the pandemic, around 14% of all cases were traceable to a single outbreak at the Netcare St Augustine Hospital in the city of Durban. This was followed by a period of exponential increase in community and household infections, signalling a localised epidemic. It is, therefore, important to hear the voices of care providers about their vulnerabilities and coping mechanisms in relation to the devastation created by COVID-19. Available research suggests that in KZN, primary healthcare practitioners experienced psychological distress, self-stigma and a disruption of the social norm [34]. Likewise, in a provincially representative study that measured the psychological impact of COVID-19 on healthcare workers, researchers found that both doctors and nurses with depression, anxiety and stress were 51%, 47.2%, and 44.3%, respectively [35]. Other KZN-based research reports that COVID-19, particularly during the lockdown period, interrupted hospital admissions [36]. Of interest to our study, in KZN, the pandemic significantly disrupted child health and subsequently impacted young people’s wellbeing [37]. Therefore, given this broad context, KZN offers a generative setting for a study like the one reported in this paper.

### 3.2. Participants and Data Generation

Twelve purposively selected male and female CYCW aged between 25–55 years participated in the study. The duration of their service in the child and youth care sector varied from 1 year to 10 years. Data was generated through semi-structured open-ended interviews that were conducted in isiZulu, a dominant language in KZN. With the participant’s permission, all the interviews were digitally recorded for analysis. Each interview session lasted 25 min. Interviews were conducted from 10 August to 18 August 2022. The interview guide included questions which were intentionally formulated for the purpose of encouraging openness in sharing experiences of working during the COVID-19 pandemic. These explored the coping mechanisms developed by CYCWs, lessons learned and overall challenges experienced. Recorded interviews were later transcribed verbatim and translated into English for analysis purposes. Participants were interviewed face-to-face, and all COVID-19 regulations were observed. Data were collected during a more relaxed regime. Hence face to face interviews were conducted. To mitigate the risk of losing or distorting the participants’ narratives, as first-language isiZulu speakers, the authors read and re-read both the isiZulu and English transcripts. Thematic analysis was used to organise the data for meaning-making.

### 3.3. Ethical Considerations

Ethical clearance to conduct this study was obtained from the Durban University of Technology’s Institutional Research Ethics Committee (041/22), while permission to access CYCW was granted by the residential CYCC authorities. Participants provided written consent for their voluntary participation in the study. To protect the identities of the participants, in this paper, we have anonymised their identities by assigning them pseudonyms.

## 4. Findings

### 4.1. COVID-19 and the Mental Health Burden on CYCW

Overall, analysis in this study revealed poor mental health as central to the participants’ narratives. They reported fear and uncertainty involving their future prospects, including anticipating negative outcomes. Fear intersected with other socioeconomic vulnerabilities and created mental health consequences for the participants. For example, a socioeconomic vulnerability reported by the participants involved the possible loss of employment and other income-generating activities. Whilst [38] found that some of the participants felt threatened with regards to working remotely because of the lack of on-hand knowledge and on-the-job training, which seems to be accurate while in the physical working environment. Further, it was revealed that COVID-19 had induced a financial crisis as people were experiencing stress due to fear and uncertainty about the extent and magnitude of the pandemic [39]. Some participants feared the prospect of retrenchment from their jobs, while others feared the possible recall of their organisational funding from external support structures. The following excerpts demonstrate:

*“I was mostly worried about my job, especially during the hard lockdown when I saw in the news that people were losing their jobs, I thought I would also experience that”*.(Agnes, Female, 45 years old)

*“We are just a non-government organisation and sometimes we hardly make money [and] that is why we always seek sponsorships to keep this organization running. So, I was scared that I would be told that I no longer have a job because some of the sponsors could not afford to support us because their businesses were affected by this pandemic”*.(Rose, Female, 30 years old)

Available research affirms these narratives of fear and uncertainty, with [40], for example, reporting on escalating psychosocial tensions resulting from the pandemic. The pandemic has got a significant psychological and social impact on the population [41], hence increasing the emotional vulnerability of front-line healthcare workers [42]. Participants also cited their fear of losing loved ones through COVID-19-related deaths. Within this context, and given that COVID-19 hits hardest on the elderly and those with a compromised immune system, participants feared the worst of the pandemic on their parents and/or older and more vulnerable members of their families. Ngcebo, for example, feared the possibility of her elderly and vulnerable mother getting infected with COVID-19 and subsequently dying from related health complications. He noted how,

*“This pandemic brought fear to me as I was scared that I might lose one of my family members, especially my mother as she is old and vulnerable, I had a bad feeling that she might get infected and not survive”*.(Ngcebo, Male, 27 years old).

Likewise, since CYCW were classified as essential service providers during the lockdown, they were thus expected to report for duty every day. This created another layer of fear, which involved the possibility of contracting COVID-19 at work and subsequently spreading it at home among household members. The pandemic contributed to the fear of infection and depression among the workforce because of the threat of close interaction with suspected cases, working long hours without getting rest and changes in sleeping and wake shapes routines [20]. Participants, such as Lindiwe, although not directly naming it, felt guilty about the prospect of infecting their family members with the virus, should they contract it during their work hours. Lindiwe explained that,

*“The thought that I always had when I was leaving home for work, was that what if I contract COVID-19 from work or anywhere in a public space and pass it to my family? That feeling caused me to think about losing my lovely family at home”*.(Lindiwe, Female, 32 years old).

Fear and uncertainty became everyday emotions that shadowed their lives, and possibly affected their work. Fear seems to have also induced other mental health-related challenges, including anxiety and stress, among the participants. For example, participants reported that working during the lockdown period, when the country saw significant spikes in reported positive COVID-19 cases, made them anxious. Anxiety resultantly affected their work, with Thobile, for example, explaining how,

*“For me, it was really difficult to do proper planning on what should I be doing with children as most of the activities were put on hold, so I was anxious about my productivity at work as it was slowly reducing”*.(Thobile, Female, 55 years old)

Other participants, such as Lindiwe, worried about the young people’s mental health under their auspices. According to this particular participant, young people in her organisation struggled to cope during the lockdown period. She said the following:

*“The only thing I was anxious about was the mental health of these young kids, I could see that some of them were barely coping with what was happening at that time and we had to teach them about this pandemic which was changing”*.(Lindiwe, Female, 32 years old).

Similarly, participants experienced work-related stress. Other participants stressed over the lives of the children they worked with, while others stressed about not being able to see their families over extended periods. The following excerpts are illustrative:

*“It was not easy to work during the hard lockdown, as the regulations were still intense. I was stressing out every time about the children I was working with because they did not understand what was going on”*.(Hector, Male, 40 years old)

*“I was not okay because I could not see my children as often as I would want to see them, you know how stressful it is to be away from your family for the longest time because were not allowed to go out as the management was worried that we would come back with COVID-19 if we went out”*.(Mbuso, Male, 25 years old)

These findings delineate the mental burden that COVID-19, and specifically the lockdown period, posed on CYCWs who were working full-time during the height of the pandemic. The findings suggest that their work was affected. It has been confirmed that in uncertain situations, such as those resulting from pandemics like COVID-19, workers’ performance becomes a concerning issue for many organizations [43]. Our findings suggest that the pandemic affected not only individuals’ health but also their physical activities, responsibilities, and daily routines.

### 4.2. Transitioning towards a “New Normal”

Another central theme that emerged from our analysis related to how CYCW adjusted to changes brought about by the pandemic. For example, South Africa’s instituted state of disaster required several non-pharmaceutical interventions to slow the spread of COVID-19. Among these requirements were strict social-physical distancing practices that governed interpersonal interactions. In response to Government’s call for safe practices, according to the participants, their work shifts changed, and they had to adhere to strict physical distancing practices when dealing with children. These changes, according to Ngcebo, were experienced as emotionally painful.

*“Our shifts changed because we had to protect ourselves from getting exposed to COVID-19. This prevented us from being in public spaces where we could contract COVID-19”*.(Ngcebo, Male, 27 years old)

*“We had to practise social distancing when we were with children and colleagues, this changed the way we used to communicate as there was no touching or hugging which was painful because we could not comfort each other and young people in the way we used to do when needed”*.(Agnes, Female, 45 years old)

Another strict measure observed during the lockdown period was an emphasis on hand hygiene. Participants had to enforce and monitor children’s hygiene practices while also continuing with their daily duties. Some participants said:

*“We had to ensure children do hygiene practice as they had to bath twice a day and at some point, they had to leave their shoes outside when they come from school so that we sanitize them properly”*.(Cath, Female, 43 years old)

*“Washing of hands regularly, avoiding touching eyes and the nose was our practice including sanitizing with alcohol-based sanitizer for hygiene purposes”*.(Beata, Female, 36 years old)

The reports suggest that the advent of the COVID-19 global pandemic has forced societies to adapt to what has been dubbed “the new normal” [44]. The term “new normal” resulted from the adaptation processes observed during the height of the COVID-19 pandemic [26]. Communities and individuals formed new habits from the learning and adaptation process, which included behavioural change [45]. Participants in this study reported similar trends in their work environment. However, while these changes were aimed at alleviating the spread of the virus, according to the participants, they challenged the nature and scope of their work. The new normal affected the participants by bringing unexpected changes to their daily routines. These enforced changes in how they engage or communicate with each other and the children and youth under their care.

## 5. Agency in Coping with COVID-19

Finally, participants demonstrated agency in coping with the emotional burden of COVID-19. They did so by actively identifying and applying specific emotionally-focused and physically-focused coping strategies by drawing on available resources. For example, an emotionally-focused coping mechanism was drawing on their spirituality through prayer. Another mechanism included engaging in interpersonal debriefing sessions among themselves as CYCW. The following excerpts illustrate:

*“The only way to deal with these challenges, I had to keep on praying for myself, family and colleagues that they are kept safe from this deadly virus”*.(Ngcebo, Male, 27 years old)

*“as CYCWs we were consulting with each other and even debriefing with each other we even grew closer to the Lord because we prayed more together”*.(Zanele, Female, 31 years old)

Indeed, participants also reported physically-focused coping strategies in the form of physical exercise. According to some of them, physical activities, such as jogging and aerobics, helped them to cope with the mental burden of COVID-19.

*“As my mental health was affected during COVID-19, I had to join my children in their daily running activity. This assisted me to deal with my mental health challenges as it made me forget about the things that severely impacted me”*.(Hector, Male, 40 years old)

*“We thought of doing aerobics with the children under our care as we were trying to entertain them which assisted me to enjoy my work in that difficult time, but we had to observe COVID-19 protocols”*.(Cath, Female, 43 years old)

While common understandings suggest that it is an arduous task to cope with pandemics, participants in this study were agentic in terms of developing coping mechanisms to curb the impact of COVID-19 and its related socioeconomic challenges. Spirituality was cited as a source of comfort and strength in coping with the pandemic [46]. Despite experiences of social distancing and having less social connectedness with friends, family members and colleagues, with spirituality, one can have faith and hope. Participants also revealed that they had to focus on physical activities, which enabled them to forget about their issues and assisted them in coping with their psychological challenges.

## 6. Limitations and Strengths

The researchers discovered limited literature that focuses on the work provided by CYCW during the COVID-19 pandemic. This means that there is limited knowledge about the narratives and experiences of CYCW. Moreover, the findings from this study must not be taken as a representation of all CYCW. Rather, we provide a snapshot of narratives provided by a small sample of CYCW, and as such, our analysis reflects these participants’ views. Whilst we have outlined the limitations of the study, we also do realise that the study findings do confirm that there are a series of weaknesses with regards to CYCCs in dealing with challenges confronted by CYCWs as there seems to be a lack of evaluation and monitoring of these issues to enable CYCWs to perform their duties effectively without having to be vulnerable to mental health issues contributed by the pandemic. Thus, systems and processes must be put in place which will adequately ensure CYCWs have support and are guided in providing essential services to children and youth in times of uncertainty and vulnerability to enhance their wellbeing. Further, the experiences shared by CYCWs do confirm that the psychological impact of COVID-19 has got uniformity across the world, including different bodies of professionals. Therefore, this strengthens evidence in terms of interventions needed to holistically address challenges within the workplace which contribute to the vulnerability of professionals. This study also adds to the limited research and literature review regarding CYCW practice, especially its importance in times of crisis.

## 7. Conclusions

The study examined the narratives of vulnerabilities and agency among a selected group of CYCW who worked during the peak of the COVID-19 global pandemic in KwaZulu-Natal, South Africa. The findings from this study confirm that CYCW experienced numerous challenges related to their duties during the pandemic. These challenges contributed to their concerns about losing their jobs, given that several organizations were rendered financially insecure, which in turn heightened their fear and uncertainty about the future. Further, fear manifested from the thought of losing loved ones as the pandemic caused numerous deaths. Hence, the above results show that some of the CYCW had psychological challenges which are associated with anxiety and stress, which could lead to depression. However, even though there were challenges experienced by the CYCW, they were able to cope as some grew their spirituality through prayer. Others saw debriefing sessions as a good coping mechanism. Moreover, others used physical activity as a coping mechanism. The findings show that COVID-19 challenged CYCW who expressed mental health-related burdens. Further, the implications of the findings suggest that CYCWs should be prepared to deal with such pandemics as they are working with vulnerable young people. Thus, CYCWs cannot appear to be vulnerable in front of young people as they should show resilience and give hope to those who have no hope. Also, the findings have implications for the mental wellbeing of CYCWs. Further, the gap that the researcher identified is that, as a developing country, the intervention of government is much needed when it comes to employee wellness, especially for those who are providing essential services in times of crisis. The issues happening during times of crisis also got some effects on their wellbeing and wellness. Thus, there is also important for policymakers to think critically about the challenges faced by essential services and the role that is needed to ensure they get supported by all parties concerned.

## Data Availability

Not applicable.

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
