# Peer review of "Vulnerability and Agency in the Time of COVID-19: The Narratives of Child and Youth Care Workers in South Africa"

_ijerph, 2023, doi:10.3390/ijerph20065010_

Round 1

Reviewer 1 Report

Vulnerability and Agency in the Time of Covid-19: The Narratives of Child and Youth Care Workers in South Africa is an important topic to be discussed in a scientific manner.  However, the paper needs to reflect more on similar studies about the influence of COVID 19 on the mental health of people in specific working environments. 

In the Introduction section, specific aims of the research and possible detected research gaps should be mentioned. Hence, this section should include previous research.

In the Materials and methods section, the interview, as the primary method used, should be more elaborated. Explain the questions used in the interview and how the questions refer to the overall topic of vulnerability and mental health...

The paper needs better formatting of the citation from the participants presented in the Findings

How are the presented findings different from general knowledge regarding the impact of COVID-19 on mental health? The presented results should be more specific in showing how COVID-19 affected the CYCW workers (other than fear or anxiety about losing their job or loved ones). The findings from the interview could be presented as a comparative analysis, or the findings could be compared to similar research studies already done on similar topics...

The paper lacks a clear aim of the study and clear contributions. 

Author Response

Dear Reviewer

I trust you are doing well. Please see attached response from authors. 

Thank you for your constructive criticism. 

Reviewer 2 Report

The text submitted for review is an example of qualitative research conducted in a specific location and on a specific small research group. Overall, the aim of the article was achieved, i.e. research was conducted to diagnose vulnerability and agency among social workers. However, I do not feel that the article adds any new knowledge to the state of existing knowledge.

Here are some shortcomings:

Firstly, there is a lack of explanation of the underlying variables: "vulnerability" and "agency" and embedding them in a theoretical perspective (psychological, sociological), which would have facilitated the analysis of the research results by making the research categories more easily distinguishable.

Secondly, the timing of the research is lacking. Was it a time of total lockdown? Or was the regime more relaxed? This is relevant to the availability of coping strategies(e.g. group physical activity outside the home).

The presentation of the results has a very simple structure: a few sentences of introduction and 2-3 citations to confirm. It seems to me that the depth of the topic is not touched upon and the analysis seems quite superficial. 

The results of the study are not surprising and rather confirm the already well-established knowledge about problems and how to deal with them in a pandemic situation. In this situation, only the place of the research itself (a child welfare institution in South Africa) is new. The question should be asked: Do the issues (both vulnerability and agency) of this specific research group differentiate it from other social groups?

The ways of coping in a pandemic situation shown in the analysis are very limited. One is puzzled by the lack of reference to family support; despite the limited contacts, this is nevertheless the closest social environment that can give much comfort and be a support in stressful situations.

There is also a lack of clear discussion with other available studies and implications for practice.

The last sentence of the text mentions implications, but only in terms of 'must' and 'should' or 'should not':

"Further, the implications of the findings suggest that CYCWs should be prepared in dealing with such pandemics as they are working with vulnerable young people, thus, CYCWs cannot appear to be vulnerable in front of the young people as they should show resilience and give hope to those who have no hope. Also, the findings have implications for the mental well-being of CYCWs."

It would be good if staff did not show vulnerability in front of children only resilience. Nevertheless, specific guidance (beyond prayer, debrifing or physical activity) would be useful. Perhaps a good guideline would be to refer to the centre's work vis-à-vis the HIV/AIDS problem and to identify good practices that will also be vital vis-à-vis COVID-19

Author Response

Dear Reviewer

I trust you are doing well. Please accept response from authors. 

Thank you for your constructive criticism. 

Round 2

Reviewer 1 Report

The paper is improved. The discussion section could be further enriched with an argument regarding the overall results and implications of similar research studies.